# Vitamin D3 Repletion Improves Vascular Function, as Measured by Cardiorenal Biomarkers in a High-Risk African American Cohort

**DOI:** 10.3390/nu14163331

**Published:** 2022-08-14

**Authors:** Satyesh K. Sinha, Ling Sun, Michelle Didero, David Martins, Keith C. Norris, Jae Eun Lee, Yuan-Xiang Meng, Jung Hye Sung, Michael Sayre, Maria Beatriz Carpio, Susanne B. Nicholas

**Affiliations:** 1Department of Medicine, David Geffen School of Medicine at University of California, Los Angeles, CA 90095, USA; 2Department of Internal Medicine, Charles R. Drew University of Medicine and Science, Los Angeles, CA 90059, USA; 3Department of Epidemiology and Biostatistics, School of Health, Jackson State University, Jackson, MS 39217, USA; 4Department of Family Medicine, Morehouse School of Medicine, Atlanta, GA 30310, USA; 5National Institute of Health, National Institute of Minority Health and Health Disparities, Bethesda, MD 20892, USA

**Keywords:** vitamin D, cardiorenal biomarker, osteopontin, FGF-23, PAI-1, African American, vascular function

## Abstract

Background: 25-hydroxy vitamin D (Vit D)-deficiency is common among patients with chronic kidney disease (CKD) and contributes to cardiovascular disease (CVD). African Americans (AAs) suffer disproportionately from CKD and CVD, and 80% of AAs are Vit D-deficient. The impact of Vit D repletion on cardio-renal biomarkers in AAs is unknown. We examined Vit D repletion on full-length osteopontin (flOPN), c-terminal fibroblast growth factor-23 (FGF-23), and plasminogen activator inhibitor-1 (PAI-1), which are implicated in vascular and kidney pathology. Methods: We performed a randomized, placebo-controlled study of high-risk AAs with Vit D deficiency, treated with 100,000 IU Vit D3 (cholecalciferol; *n* = 65) or placebo (*n* = 65) every 4 weeks for 12 weeks. We measured kidney function (CKD-EPI eGFR), protein-to-creatinine ratio, vascular function (pulse wave velocity; PWV), augmentation index, waist circumference, sitting, and 24-h-ambulatory blood pressure (BP), intact parathyroid hormone (iPTH) and serum calcium at baseline and study end, and compared Vit D levels with laboratory variables. We quantified plasma FGF-23, PAI-1, and flOPN by enzyme-linked immunosorbent assay. Multiple regression analyzed the relationship between log flOPN, FGF-23, and PAI-1 with vascular and renal risk factors. Results: Compared to placebo, Vit D3 repletion increased Vit D3 2-fold (*p* < 0.0001), decreased iPTH by 12% (*p* < 0.01) and was significantly correlated with PWV (*p* < 0.009). Log flOPN decreased (*p* = 0.03), log FGF-23 increased (*p* = 0.04), but log PAI-1 did not change. Multiple regression indicated association between log flOPN and PWV (*p* = 0.04) and diastolic BP (*p* = 0.02), while log FGF-23 was associated with diastolic BP (*p* = 0.05), and a trend with eGFR (*p* = 0.06). Conclusion: Vit D3 repletion may reduce flOPN and improve vascular function in high risk AAs with Vit D deficiency.

## 1. Introduction

Twenty-five hydroxy vitamin D (Vit D) is a fat-soluble vitamin which is hydroxylated in the liver and converted to 1,25-dihydroxy vitamin D (1,25D), in the renal proximal tubule [1]. The classical function of 1,25D is to regulate calcium and phosphate homeostasis and bone mineralization, but it also plays important roles in glucose metabolism, vascular function, cardiomyocyte health, immunity and inflammation [2,3,4,5]. Approximately 30–50% of the general population are Vit D deficient [6]. Importantly, Vit D deficiency may contribute to the pathogenesis of numerous diseases, including chronic kidney disease (CKD), and cardiovascular disease (CVD) [2,5,7,8].

Although, the prevalence of Vit D deficiency (<30 ng/mL) has been reported to be 30–50% in the general population [9], African Americans (AAs) have a prevalence of Vit D deficiency as high as 90% [10] and have constantly had lower serum Vit D levels compared to their White peers [9]. Notably, AAs also suffer a higher prevalence of CKD and CVD, that may be related, in part, to their lower serum Vit D levels [10,11,12]. Furthermore, it is becoming increasingly apparent that normal Vit D levels protect against several chronic conditions, including both CKD and CVD, which are also highly prevalent among AAs compared to their White peers [13,14]. Vit D deficiency can be effectively managed with oral supplementation and/or modest sunlight exposure [15] and, interestingly, Vit D supplementation/repletion has been demonstrated to improve arterial stiffness [16] and glycemic control [17], inhibit renin-angiotensin system [18], and preserve renal function [19], as well as lead to significant favorable changes in serum inflammatory markers [17].

In the current study, we tested the hypothesis that Vit D supplementation in a Vit D deficient AA cohort with controlled hypertension and preserved kidney function may improve clinical and biochemical indicators of cardiovascular and kidney pathology. To demonstrate this, we quantified a number of biomarkers that have been implicated in the pathology of CVD and CKD: full length osteopontin (flOPN), c-terminal fibroblast growth factor-23 (FGF-23), and plasminogen activator inhibitor-1 (PAI-1), to assess the effects of Vit D repletion after 12 weeks on these established indicators of vascular and kidney function. In particular, flOPN is a multifunctional, ubiquitously expressed adhesion protein that plays a pivotal role in coexistent cardiometabolic disorders including obesity, CKD, and CVD [20,21]. Further, flOPN may modulate angiotensin-II-induced fibrosis in the heart [22] and kidney [23] and its deficiency prevents the development of diabetic nephropathy [20]. FGF-23 is a bone-derived hormone that suppresses phosphate reabsorption and the synthesis of 1,25D in the kidney. FGF-23 may also be an early biomarker for kidney dysfunction as well as a predictor for CVD risk and mortality in patients with CKD [24]. PAI-1 belongs to the superfamily of serine protease inhibitor serpin and prevents the conversion of tissue plasminogen activator and urokinase plasminogen activator to plasminogen. As such, PAI-1 regulates fibrinolysis, thrombosis, fibrosis and tissue remodeling, and its genetic deficiency has been shown to prevent diabetic nephropathy [25].

## 2. Materials and Methods

### 2.1. Study Population and Design

For this study 737 participants were screened between September 2009 to August 2011 at affiliated medical clinics at Charles R. Drew University and Morehouse School of Medicine [26,27]. A total of 607 participants were excluded as they did not meet the inclusion criteria: hypertensive with serum Vit D between 10 ng/mL and 25 ng/mL. As previously described [27], participants were excluded if they had poorly controlled hypertension (systolic BP > 160 mm Hg or diastolic BP > 100 mm Hg), severe Vit D deficiency (25(OH)D < 10 ng/mL), diabetes mellitus (DM; fasting blood glucose (FPG) > 125 mg/dL or HbA1c > 6.5%), hypercalcemia (serum calcium > 10.5 mg/dL), CKD (2009 CKD-EPI estimated glomerular filtration rate (eGFR) [28] < 45 mL/min/1.73 m^2^), stroke or congestive heart failure, history of kidney stones, allergy to cholecalciferol or microcrystalline cellulose, abnormal liver function tests, recent (<6 months) hospitalization for myocardial infarction, used immunosuppressive, chronic steroid therapy, or non-steroidal anti-inflammatory drugs [27]. Race and ethnicity were self-reported. Finally, at each site, 65 male and female AAs participants (18–70 years old) were recruited. The study was approved by the local institutional review boards, all participants signed a written informed consent and the study was registered with ClinicalTrials.gov (accessed on June 6, 2022, Identifier: NCT02802449).

In addition to a brief medical history, a limited physical examination measured height (cm), weight (kg), and waist circumference (WC), as well as calculated body mass index (BMI). Participants were randomized to receive a monthly dose of 100,000 IU of cholecalciferol [26] or a placebo for three months. Participants were followed and blood and urine were collected for analyses at baseline and 12 weeks.

At baseline and week 12 visits, blood pressure (BP; an average of three seated readings), vascular function, endothelial function, and pulse wave velocity (PWV) were assessed. The function of the vasculature was evaluated using radial artery tonometry and computer-aided measurement of the augmentation index (AI; the ratio of arterial augmentation pressure to arterial pulse pressure) using the SphygmoCor product family, which is designed to measure the pressure wave in the ascending aorta from an external measurement. In order to assess endothelial function, the Endopat noninvasive vascular finger device was used to measure percent flow-mediated dilation [29].

### 2.2. Enzyme Linked Immunosorbent Assays (ELISAs)

Plasma levels of flOPN, C-terminal FGF-23, and PAI-1 were measured using the commercially available ELISA kits. The flOPN kits were purchased from Immuno-Biological Lab America (cat# 27158 (Immuno-Biological Lab America: Minneapolis, MN, USA); sensitivity 3.33 ng/mL, measurement range 0.07~4.75 ng/mL) and the manufacturer’s instructions were followed to perform the ELISA. Briefly, 100 μL each of test samples, diluted standard, reagent blank, and test sample blank was put into the pre-coated wells. The plate was incubated for 1 h at 37 °C and washed four times. After washing, we used 100 μL of labeled antibody into the wells of test samples, diluted standard, and test sample blank. The plate was incubated for 30 min at 4 °C and washed five times with prepared wash buffer. Then, 100 μL chromogen was used in the wells and the plate was incubated for 30 min at room temperature in the dark and 100 μL of stop solution was used to stop the reaction. The absorbance was measured at 450 nm.

The kits for FGF-23 were purchased from QUIDEL (Cat# 60–6100 (QUIDEL:San Diego, CA, USA); sensitivity 1.5 RU/mL, measurement range 19.4–1455 RU/mL). The assay was performed following the manufacturer’s protocol. In summary, 100 μL of standard, control, or sample were added to the designated wells and working antibody solution (50 μL) was added to each well. The plate was incubated at room temperature for three hours on a horizontal rotator set at 180–220 RPM and was washed five times. Then, 150 μL of ELISA HRP Substrate was added into each well and the plate was incubated at room temperature for 30 min on a horizontal rotator. The absorbance was taken at 620 nm within 5 min against the 0 RU/mL standard wells as a blank. Immediately 50 μL of stop solution was added into each of the wells and was mixed on a horizontal rotator for 1 min. The absorbance at 450 nm within 10 min was read against a reagent blank.

The PAI-1 ELISA kits were purchased from Fischer Scientific (Cat# KHC3071 (Fisher Scientific, Hampton, NH, USA); sensitivity < 30 pg/mL, measurement range 31.3–2000 pg/mL) and assay was performed according to the manufacturer’s instructions. Briefly, 100 µL of standards, pre-diluted samples or controls were added to the appropriate wells and the plate was incubated for two hours at room temperature. The plate was washed 4 times with wash buffer. Then, 100 µL Hu PAI-1 biotin conjugate solution was added into each well, except the chromogen blanks. The plate was incubated for 2 h at room temperature and was washed 4 times. 100 µL of 1X streptavidin-HRP solution was added to each well except the chromogen blanks. The plate was incubated for 30 min at room temperature then washed and 100 µL of stabilized chromogen was added to each well. After 30 min of incubation at room temperature in the dark, 100 µL stop solution was added to each well. The absorbance was measured at 450 nm.

### 2.3. Statistics

All analyses were performed using SAS version 9.2 (SAS Institute Inc., Cary, NC, USA) or JMP pro 16 (SAS Institute Inc., Cary, NC, USA,). We compared baseline characteristics between the two groups using Chi-Square tests for categorical variables and *t*-tests for continuous variables Serum flOPN, FGF-23, and PAI-1 levels were not normally distributed and therefore were log-transformed (Appendix A). Multivariable regression assessed the relationship between Vit D and participant variables (demographic, physiologic and laboratory). To analyze the efficacy of the intervention, we used the maximum-likelihood mixed-effects repeated-measures model. Tests were conducted to determine whether there was a difference between study groups in the impact of Vit D3 supplementation on serum Vit D by analyzing the interaction term between the change of the endpoints and study arms. Biomarkers’ dependent changes in independent variables from baseline to week 12 were assessed using multiple regression analysis. A covariate of site identification was included in all statistical models to control for potential differences between sites. Multiple imputation, however, is better suited to longitudinal studies with missing values, as it is more powerful than the mixed model without imputation. Accordingly, we compared the results from both methods and found no significant differences. Statistical significance was set at *p* < 0.05, and the t-value was the calculated difference in units of standard error.

## 3. Results

Baseline demographic, physiologic and clinical data were reported as part of a prior analysis of this cohort to assess the association of Vit D3 supplementation and both inflammatory and oxidative mediators of arterial stiffness [27]. Briefly, study participants had a mean age of 50.0 ± 9.5 years, 61% were female, mean BMI was 34.5 ± 5.2 kg/m^2^, mean WC was 104.7 ± 12.4 cm, mean BP was 127 ± 16/83 ± 11 mm Hg, PWV was 9.1 ± 2.2 m/s, mean augmentation pressure was 11.7 ± 6.6 mm Hg, mean serum calcium (9.32 ± 0.3 mg/dL) and AI was 29.8 ± 11.4% (Table 1) with no significant differences between the placebo and treated groups. In addition, Vit D (16.8 ± 5.1 ng/mL) and iPTH (45.5 ± 24.7 pg/mL) levels did not differ between groups. Although baseline eGFR was slightly lower in the Vit D3 treated group compared to placebo subjects (96 ± 18.2 vs. 103.7 ± 17.6 mL/min/1.73 m^2^, *p* = 0.02), Vit D levels did not differ between placebo and Vit D3-treated groups.

The monthly cholecalciferol dose of 100,000 IU was associated with a 2-fold increase in serum 25(OH)D from 17 ± 5 ng/mL to 35 ± 7 ng/mL (*p* < 0.0001) after 12 weeks. The increase in the serum Vit D levels was associated with an ~12% decrease in serum iPTH (*p* < 0.01, Table 2a), and no change in other physiologic parameters, including urine PCR. However, the serum levels of log transformed flOPN was significantly reduced (*p* = 0.03) and log FGF-23 was increased in the treated group (*p* = 0.04) but no change was observed in log PAI-1, Table 2a. There was a significant correlation between Vit D and PWV and BMI in the bivariate model for all measured parameters, that persisted for PWV in the fully adjusted multivariate regression model (Table 2b).

To examine the effect of the strength of the relationship between Vit D repletion between baseline and week 12, the observed decrease in log flOPN, and the increase in log FGF-23, multiple regression analyses were performed. Table 3a shows that the decrease in log flOPN was negatively correlated with PWV (*p* = 0.04), and positively associated with diastolic BP (*p* = 0.02). The increase in log FGF-23 between baseline and week 12 was positively associated with a trend in eGFR (*p* = 0.06), and diastolic BP (*p* = 0.05) (Table 3b).

## 4. Discussion

To our knowledge, this is the first evidence of a direct effect of Vit D3 repletion on circulating levels of cardiorenal biomarkers that have been implicated in cardiac and kidney pathology. We showed that Vit D3 repletion correlated with lower PWV. In addition, lower flOPN correlated with lower PWV and higher diastolic BP, and higher FGF-23 levels correlated with higher diastolic BP, while Vit D3 repletion had no impact on PAI-1 levels. PWV is a measure of arterial stiffness, and is a strong and independent predictor of future CV events and all-cause mortality [30]. Notably, recent studies have demonstrated the positive association of OPN with PWV to predict aortic stiffness in hypertensive [31] and peritoneal dialysis patients [32]. OPN has also been shown to be associated with vascular function in patients with coronary artery disease [33], thus linking OPN with CVD through arterial wall stiffening. Diastolic BP is a strong determinant of arterial stiffness and CV risk [34]. In fact, early studies showed that diastolic BP, more than systolic BP, predicted coronary heart disease, particularly in younger and middle-aged individuals [35]. The effects of Vit D3 supplementation on CV risk factors and hypertension [36] have been widely explored, but conflicting results remain. It is known that AAs have a higher prevalence of hypertension than other racial and ethnic groups [37]. From our results, it is likely that flOPN may be a marker of vascular function in AAs deficient of Vit D.

Interestingly, the ability for flOPN to modify vascular injury has also been explored [38]. In particular, flOPN-meditated inflammation promotes vascular remodeling in atherosclerosis [38], whilst flOPN may possess protective effects against vascular calcification [39]. Although, there is no clear-cut, known mechanism through which Vit D may affect flOPN level, Vit D3 is reported to induce skeletal anabolism and couples the activity of osteoblasts and osteoclasts through the regulation of several genes, including flOPN [40,41]. One study indicated that circulating levels of flOPN were closely related to GFR and CV risk markers in patients with CKD [42]. Elevated plasma flOPN was also reported in a cohort of patients with CKD stage 5 receiving hemodialysis (CKD 5-HD) compared to a normal healthy cohort [43]. Further, a significant positive correlation of flOPN with iPTH and alkaline phosphatase was observed in CKD 5-HD patients [43]. Another study indicated that flOPN expression in aortic vascular smooth muscle cells was increased by Vit D receptor agonist treatment, in contrast to circulating flOPN levels that decreased in conjunction with reduced vascular calcification, explaining its local action [44].

FGF-23 is an emerging novel and powerful risk factor of mortality and CV events in patients with CKD and end stage renal disease [45]. Studies that have examined the effect of Vit D3 on FGF-23 have shown conflicting results. We reported an increase in log FGF-23 with improved eGFR that did not reach statistical significance. A previous study indicated no significant effect of Vit D3 supplementation on FGF-23 level [46] while Charoenngam et al. observed an increase in serum C-terminal FGF-23 concentration [47]. Another study identified significant interactions among Vit D, Vit D3, and FGF-23 on cardiac remodeling showing increased left ventricular mass and cardiac dilatation with low Vit D and high FGF-23 [48]. Together with Vit D3 and iPTH, FGF-23 is part of a complex multi-tissue feedback system in regulating calcium and phosphate homeostasis [49]. By reducing the expression of sodium-phosphate cotransporters (Npt2a and Npt2c), FGF-23, in conjunction with its cofactor klotho, exert phosphaturic effects on the kidney. It also impairs the formation of Vit D3 and decreases iPTH mRNA in the parathyroid glands [50]. Since Vit D3 supplementation leads to a suppression of iPTH and an increase in Vit D3 [51,52], a resulting rise in serum phosphate levels may in turn stimulate the synthesis of FGF-23 [45]. Thus, we can speculate that the small rise in FGF-23 in our study may be regarded as a defensive mechanism against Vit D-induced phosphate increases.

We also observed that Vit D3 supplementation for 12 weeks was sufficient to change the Vit D level by 100% with a significant reduction in iPTH. This finding is consistent with a previous study where lower levels of Vit D were reported to be associated with higher levels of serum iPTH in the absence of kidney disease [53]. Through utilization of the National Health and Nutrition Examination Study (NHANES), Gutierrez and colleagues conducted the largest assessment of Vit D and iPTH in a diverse population, including White, Hispanic/Latinx, and AA study participants [54]. AA and Hispanic/Latinx participants had significantly lower Vit D and higher iPTH concentrations than White participants and the association of Vit D deficiency (≤20 ng/mL) and iPTH levels was modified by race and ethnicity. Elevated levels of iPTH were found to be associated with an increased risk for CVD and mortality in patients with CKD [55]. A significant decrease in mortality and heart failure was observed by pharmacologic lowering of elevated iPTH levels in patients with CKD [56]. In CKD, iPTH levels rise as soon as eGFR falls below 60 mL/min/1.73 m^2^. Phosphate retention, low plasma calcium, and high FGF-23 with subsequently decreased plasma Vit D concentration, all contribute to the involvement of secondary hyperparathyroidism [57,58].

The strengths of this study are that it was a randomized controlled trial and demonstrated the well-proven therapeutic effect of Vit D3 repletion on 25 (OH) D and iPTH [45,59]. Most importantly, for the first time, we have observed flOPN to be reduced by Vit D3 supplementation that may improve vascular function and overall vascular health. Therefore, our results have significantly added to the existing data in this area. The limitations of the current study are the degree of statistical significance and sensitivity to smaller effect sizes due to the relatively small population size at two sites.

## 5. Conclusions

In conclusion, our findings indicate that Vit D3 repletion may improve vascular function in high risk AA study participants with Vit D deficiency, obesity and controlled hypertension, and that flOPN could be a more sensitive marker of vascular function, compared to FGF-23 in this population, with no significance related to PAI-1. Further study is warranted in other larger, geographically diverse, racial and ethnic populations to validate our findings and to increase generalizability.

## Figures and Tables

**Table 1 nutrients-14-03331-t001:** Baseline characteristics for control and vitamin D3 treated groups.

	Total	Placebo Control *n =* 65	Vitamin D Treated *n =* 65	*p*-Value
**Demographic characteristics**				
Age, years, mean (SD)	50.0 (9.5)	49.5 (8.8)	50.4 (10.1)	0.60
Sex, N (%)				
Male/Female	51/79 (39)	27/38 (42)	24/41 (37)	0.59
Body mass index, kg/m^2^	34.5 (5.2)	34.30 (5.6)	34.7 (4.8)	0.64
Waist circumference, cm	104.7 (12.4)	104.4 (12.5)	105 (12.4)	0.80
**Physiologic characteristics, mean (SD)**				
CKD-EPI eGFR (mL/min/1.73 m^2^)	100.2 (18.2)	103.7 (17.6)	96.8 (18.2)	0.02
Urine protein to creatinine ratio (mg/g)	4.8 (6.4)	4.7 (6.0)	4.9 (6.8)	0.97
Serum 25(OH) D3 (ng/mL)	16.8 (5.1)	16.5 (5.0)	17.0 (5.2)	0.60
Intact PTH (pg/mL)	45.5 (24.7)	50 (34)	43.0 (20)	0.20
Serum Calcium	9.32 (0.3)	9.30 (0.4)	9.35 (0.3)	0.44
Log FGF-23	4.31 (0.7)	4.34 (0.8)	4.3 (0.6)	0.70
Log PAI-1	7.14 (0.7)	7.3 (0.6)	7.0 (0.8)	0.12
Log flOPN	5.06 (1.0)	4.9 (1.2)	5.3 (0.7)	0.05
**Cardiac characteristics, mean (SD)**				
Systolic BP (mm Hg)	127 (16)	128.5 (15.2)	125.4 (16.1)	0.22
Diastolic BP (mm Hg)	83 (11)	84.5 (10.5)	81.1 (11.4)	0.10
24-h systolic BP (mm Hg)	128.2 (13.1)	130.0 (13.5)	126.7 (12.6)	0.15
24-h diastolic BP (mm Hg)	78.4 (8.5)	79.5 (8.8)	77.4 (8.2)	0.60
Pulse wave velocity (m/s), mean (SD)	9.1 (2.2)	9.1 (1.8)	9.1 (2.4)	0.90
Augmentation pressure (mm Hg)	11.7 (6.6)	12.1 (6.8)	11.4 (6.4)	0.60
Augmentation index (%)	29.8 (11.4)	31.0 (12)	28.1 (11.2)	0.20

FGF-23: c-terminal fibroblast growth factor-23; PAI-1: plasminogen activator inhibitor-1; flOPN: full length osteopontin; CKD-EPI eGFR: Chronic Kidney Disease-Epidemiology Collaboration equation to estimate glomerular filtrate rate; PTH: parathyroid hormone.

**Table 2 nutrients-14-03331-t002:** (**a**) Change in patient characteristics from baseline and week 12 for placebo control and Vitamin D treated groups. (**b**) Multivariate regression analysis of the relationship between Vit D and participant variables, at week 12.

(a)						
	Placebo Control Mean (SD)			Vit D Treated Mean (SD)		
	Baseline	Week 12	*p*-Value	Baseline	Week 12	*p*-Value
**Demographic characteristics**						
Body mass index, kg/m^2^	34.30 (5.6)	34.30 (5.6)	1.00	34.7 (4.8)	34.7 (4.8)	1.00
Waist circumference, cm	104.4 (12.5)	104.4 (12.5)	1.00	105 (12.4)	105 (12.4)	1.00
**Physiologic characteristics, mean (SD)**						
CKD-EPI eGFR (mL/min/1.73 m^2^)	103.7 (17.6)	99.1 (16.9)	0.15	96.8 (18.2)	96.1 (19.7)	0.84
Protein to creatinine ratio (mg/g)	4.7 (6.0)	4.6 (4.24)	0.74	4.9 (6.8)	4.4 (5.3)	0.99
Serum 25(OH) D (ng/mL)	17 (5)	17 (6)	0.53	17 (5)	35 (7)	<0.0001
Intact PTH (pg/mL)	50 (34)	50 (38)	0.31	43 (20)	38 (16)	<0.01
Serum Calcium	9.3 (0.4)	9.29 (0.4)	0.84	9.35 (0.3)	9.39 (0.3)	0.46
Log FGF-23	4.3 (0.8)	4.5 (0.6)	0.67	4.3 (0.6)	4.5 (0.5)	0.04
Log PAI-1	7.3 (0.6)	7.2 (0.8)	0.67	7.0 (0.8)	7.1 (0.8)	0.84
Log flOPN	4.9 (1.2)	5.0 (1.2)	0.59	5.3 (0.7)	4.9 (1.3)	0.03
**Cardiac characteristics, mean (SD)**						
Systolic blood pressure (mm Hg)	128.5 (15.2)	125.8 (13.4)	0.67	125.4 (16.1)	126.9 (15)	0.69
Diastolic blood pressure (mm Hg)	84.5 (10.5)	82.2 (9.2)	0.38	81.1 (11.4)	81.1 (12)	0.68
24 h systolic blood pressure (mm Hg)	130.0 (13.5)	130.8 (13.2)	0.81	126.7 (12.6)	128.3 (15.2)	0.61
24 h diastolic blood pressure (mm Hg)	79.5 (8.8)	79.2 (10.3)	0.88	77.4 (8.2)	78 (11)	0.76
Pulse wave velocity (m/s)	9.1 (1.8)	9.1 (2.0)	0.96	9.1 (2.4)	8.9 (2.3)	0.92
Augmentation pressure (mmHg)	12.1 (6.8)	11.0 (5.4)	0.64	11.3 (6.4)	12.5 (14.6)	0.51
Augmentation index (%)	31.0 (12)	29.3 (11.1)	0.73	28.1 (11.2)	27.6 (11.1)	0.92
(**b**)						
**Independent Variables**	**Parameter Estimate**		**SE**		**t-Value**	***p*-Value**
**Bivariate Correlation analysis**						
PWV	0.09		0.04		2.12	0.04
BMI	0.23		0.09		2.75	0.01
**Multivariate regression analysis ***						
PWV	1.31		0.48		2.73	0.01
BMI	0.40		0.34		1.18	0.23

FGF-23: c-terminal fibroblast growth factor-23; PAI-1: plasminogen activator inhibitor-1; flOPN: full length osteopontin; CKD-EPI eGFR: Chronic Kidney Disease-Epidemiology Collaboration equation to estimate glomerular filtrate rate; PTH: parathyroid hormone. * Multivariate regression analysis controlling for waist circumference, systolic and diastolic blood pressure, augmentation pressure, augmentation index, eGFR, protein to creatinine ratio, intact PTH, and serum calcium. PWV: pulse wave velocity; BMI: body mass index.

**Table 3 nutrients-14-03331-t003:** (**a**,**b**) Multiple regression analysis of log flOPN, and FGF-23 showing dependent change of log-biomarker from baseline to week 12 and independent variables.

**(a)**					
**Dependent Variable**	**Independent Variable**	**Parameter Estimate**	**S.E.**	**t-Value**	***p*-Value**
Change of log flOPN between baseline and week 12	CKD-EPI eGFR (mL/min/1.73 m^2^)	0.009	0.008	1.09	0.28
	Protein creatinine ratio (mg/g)	−0.042	0.080	−0.52	0.60
	Pulse wave velocity (m/s)	−0.169	0.082	−2.05	0.04
	Augment Pressure (mm Hg)	−0.007	0.021	−0.31	0.75
	Augmentation Index (%)	0.009	0.012	0.69	0.50
	Systolic BP (mm Hg)	0.003	0.009	0.27	0.79
	Diastolic BP (mm Hg)	0.032	0.014	2.33	0.02
	24-h Systolic BP (mm Hg)	−0.014	0.012	−1.12	0.27
	24-h Diastolic BP (mm Hg)	−0.006	0.018	−0.34	0.73
**(b)**					
**Dependent Variable**	**Independent Variable**	**Parameter Estimate**	**S.E.**	**t-Value**	***p*-Value**
Change of Log FGF-23 between baseline and week 12	CKD-EPI eGFR (mL/min/1.73 m^2^)	0.005	0.003	1.87	0.06
	Protein creatinine ratio (mg/g)	0.010	0.028	0.36	0.72
	Pulse wave velocity (m/s)	−0.021	0.027	−0.80	0.43
	Augmentation pressure (mm Hg)	−0.012	0.008	−1.58	0.12
	Augmentation Index (%)	−0.004	0.004	−0.85	0.40
	Systolic BP (mm Hg)	0.003	0.003	0.95	0.34
	Diastolic BP (mm Hg)	0.010	0.005	1.99	0.05
	24-h Systolic BP	0.006	0.005	1.18	0.24
	24-h Diastolic BP	0.0100	0.008	1.19	0.24

Multiple regression analysis controlling for placebo group; CKD-EPI eGFR: Chronic Kidney Disease-Epidemiology Collaboration equation to estimate glomerular filtrate rate; BP: blood pressure; flOPN: full-length osteopontin; FGF-23: fibroblast growth factor-23.

## Data Availability

The data presented in this study are available upon request from the corresponding author.

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
