# Peer review of "Vitamin D3 Repletion Improves Vascular Function, as Measured by Cardiorenal Biomarkers in a High-Risk African American Cohort"

_nutrients, 2022, doi:10.3390/nu14163331_

Round 1
Reviewer 1 Report
Dear Editor,
Thank you very much for an opportunity to review the manuscript nutrients-1783498. This is very elegant study aiming to elucidate the impact of vitamin D deficiency on cardiovascular system and kidney function. This manuscript could be of high interest to Nutrients reader after some clarifications.
I have some comments:
1. I am aware that it was interventional study, but still you could try to plot all clinical (biochemical) parameters against 25(OH)D level and see if there are any correlations.
2. “Serum Vit D, iPTH and calcium levels at 4, 8 and 12 weeks”, why are you not presenting these data, it could be of interest to see dynamics of 25(OH)D levels. I wonder, whether 25(OH)D levels differ after 4, 8 and 12 weeks?
3. Did you measure classic calcium-phosphate parameters?Anything interesting?
4. Title “rRepletion”
5. Author Contributions section was not completed
Reviewer 2 Report
Sinha et conducted RCT of vitamin D supplementation or placebo for Cardiorenal Biomarkers in AA cohort. This study is interesting and meaningful for patients with CKD and nephrologist.
The study design is relatively appropriate and manuscript is well written.
On the other hand, this study contains important problems in study design and manuscript in method section.
1. The author had measured 25OHD, FGF23, PAI-1, and OPN as a biomarker study for cardiovascular disease. On the other hand, I think the markers with evidence of conventional cardiovascular disease had not been measured. I think the author should measure evidenced markers such as BNP, troponin, LVH in UCG, and cacs.
ï¼’.There is a description of 25ohd, pth, etc. after 12 weeks, but there is no description of Ca, P, etc. Also, the concentration of active vitamin D has not been measured. Nutritional status and inflammatory status are important factors that affect vitamin D, but they are not described in this study.
3.
I think it would be better to have a description of the research period, how to determine the number of cases, and whether or not withdrawal after allocation. In addition, allocation method was not described in the manuscript.
In the baseline patient background, it would be helpful to have data on medications such as the use of antihypertensive drugs and the presence or absence of VitD supplementation, and information on the presence or absence of smoking.
Prior to this study, the relationship between flOPN and PWV / diastolic blood pressure and the relationship between FGF-23 and diastolic blood pressure shown in this study were described, and various factors including cardiovascular events and arteriosclerosis were described. There was a previous report showing the correlation, and I felt that the logic was a little leap to show that flOPN is a marker of vascular function of AA.
